# Predicting Tumor Dynamics Post-Staged GKRS: Machine Learning Models in Brain Metastases Prognosis

**DOI:** 10.3390/diagnostics14121268

**Published:** 2024-06-15

**Authors:** Ana-Maria Trofin, Călin Gh. Buzea, Răzvan Buga, Maricel Agop, Lăcrămioara Ochiuz, Dragos Teodor Iancu, Lucian Eva

**Affiliations:** 1University of Medicine and Pharmacy “Grigore T. Popa” Iași, 700115 Iasi, Romania; ana-maria.trofin@umfiasi.ro (A.-M.T.); lacramioara.ochiuz@umfiasi.ro (L.O.); dt_iancu@yahoo.com (D.T.I.); 2Clinical Emergency Hospital “Prof. Dr. Nicolae Oblu” Iași, 700309 Iasi, Romania; calinb2003@yahoo.com (C.G.B.); elucian73@yahoo.com (L.E.); 3National Institute of Research and Development for Technical Physics, IFT Iași, 700050 Iasi, Romania; 4Physics Department, Technical University “Gheorghe Asachi” Iasi, 700050 Iasi, Romania; m.agop@yahoo.com; 5Regional Institute of Oncology, 700483 Iasi, Romania; 6University Apollonia, 700511 Iasi, Romania

**Keywords:** gamma knife radiosurgery (GKRS), brain metastasis, tumor dynamics forecasting, machine learning models, feature importance

## Abstract

This study assesses the predictive performance of six machine learning models and a 1D Convolutional Neural Network (CNN) in forecasting tumor dynamics within three months following Gamma Knife radiosurgery (GKRS) in 77 brain metastasis (BM) patients. The analysis meticulously evaluates each model before and after hyperparameter tuning, utilizing accuracy, AUC, and other metrics derived from confusion matrices. The CNN model showcased notable performance with an accuracy of 98% and an AUC of 0.97, effectively complementing the broader model analysis. Initial findings highlighted that XGBoost significantly outperformed other models with an accuracy of 0.95 and an AUC of 0.95 before tuning. Post-tuning, the Support Vector Machine (SVM) demonstrated the most substantial improvement, achieving an accuracy of 0.98 and an AUC of 0.98. Conversely, XGBoost showed a decline in performance after tuning, indicating potential overfitting. The study also explores feature importance across models, noting that features like “control at one year”, “age of the patient”, and “beam-on time for volume V1 treated” were consistently influential across various models, albeit their impacts were interpreted differently depending on the model’s underlying mechanics. This comprehensive evaluation not only underscores the importance of model selection and hyperparameter tuning but also highlights the practical implications in medical diagnostic scenarios, where the accuracy of positive predictions can be crucial. Our research explores the effects of staged Gamma Knife radiosurgery (GKRS) on larger tumors, revealing no significant outcome differences across protocols. It uniquely considers the impact of beam-on time and fraction intervals on treatment efficacy. However, the investigation is limited by a small patient cohort and data from a single institution, suggesting the need for future multicenter research.

## 1. Introduction

Brain metastases (BMs), often referred to as secondary brain cancers, pose significant treatment dilemmas and urgently need strategies to lessen the impact on those diagnosed [1]. There’s an increasing incidence of BMs, likely influenced by improvements in conventional therapies like surgery, radiation, and chemotherapy, which have led to longer survival rates for patients. Without any medical interventions, people with BMs usually face a median survival time of approximately 2 months after their diagnosis, especially when the disease affects the central nervous system [2]. The advent of targeted treatment methods, such as Gamma Knife radiosurgery (GKRS), is becoming a preferred approach for BM management due to its ability to better target tumors locally with fewer side effects compared to traditional whole brain radiation therapy, and its effectiveness is on par with that of surgical removal [3,4,5,6]. However, GKRS is limited in treating larger BMs (exceeding 3 cm in diameter or 10 cc in volume) because of the potential for radiation-induced harm, like radiation toxicity [7,8,9,10]. The Gamma Knife ICON, using mask fixation for patient positioning, represents an advanced solution for hypo-fractionated treatment of large BMs. Despite these advancements, the median survival rate for patients with BMs hovers around one year [11,12].

Lung cancer stands as a major global cancer threat, categorized mainly into non-small cell lung cancer (NSCLC) and small cell lung cancer. NSCLC is further divided into adenocarcinoma, squamous cell carcinoma, and large cell carcinoma, as classified by the World Health Organization [13,14]. Between 16% and 60% of non-small cell lung cancer (NSCLC) patients develop brain metastases (BMs), irrespective of whether their disease is driven by oncogenes or lacks actionable mutations [15,16]. Addressing BMs in NSCLC requires an integrated approach to achieve timely control of the disease. Unfortunately, the effectiveness of standard treatments such as surgery, radiotherapy, and corticosteroid palliation is limited, and these treatments are often associated with severe neurotoxic effects that can delay or impair systemic therapy, leading to high mortality rates [17,18,19,20]. The introduction of advanced neuro-imaging techniques and more effective systemic therapies has extended the lives of NSCLC patients, thereby increasing the overall incidence of BMs during their lifetime [21]. Treatment options like stereotactic radiosurgery (SRS), either as a standalone treatment or following surgery and whole-brain radiotherapy (WBRT) for specific cases is considered [22], although they carry the risk of significant neurocognitive side effects. The use of GKRS for NSCLC patients with BMs is debated due to their limited OS. Nevertheless, some policies, like those implemented in Korea, permit the use of GKRS for BM treatment every three months.

Beyond lung cancer, other cancers such as breast, colorectal, prostate, ovarian, renal cancers, and melanoma also metastasize to the brain, significantly impacting patient outcomes and treatment approaches. Breast cancer, for example, has a notable propensity to spread to the brain, particularly in patients with HER2-positive or triple-negative subtypes, challenging clinicians to tailor treatments for both systemic disease and brain involvement [23]. Colorectal cancer, though less commonly associated with BM, poses a significant risk when metastasis occurs, necessitating a multidisciplinary approach to manage both the primary disease and brain metastases [24]. Prostate cancer rarely metastasizes to the brain, but when it does, it signifies a late-stage disease and poor prognosis, highlighting the need for innovative therapeutic strategies [25,26]. Ovarian and renal cancers, similarly, can lead to brain metastases, with renal cell carcinoma being more prone to spread to the brain, requiring careful consideration of treatment options to address this aggressive disease behavior [27,28]. Brain metastases (BMs) occur in almost 50% of patients with cutaneous melanoma (CM) and are the third most common metastatic site [29].

Machine learning (ML), as a branch of artificial intelligence, is instrumental in crafting models that learn from data autonomously, without direct programming. Among these, tree-based ML algorithms are particularly appreciated for their straightforwardness and clarity, making the decision paths visual and interpretable. Decision trees stand out by mapping decisions through nodes and labels, ensuring not only high accuracy in classification but also clear presentation of information, which is crucial in healthcare decision-making [30,31]. However, challenges such as overfitting can arise from small data sets. Techniques like random forests and boosted decision trees are therefore employed to overcome these hurdles, offering improved prediction accuracy through a detailed examination of the interconnections among variables in the dataset [32].

In cases where the data consist of clinical features, deep learning (DL) can be highly beneficial, particularly in uncovering intricate relationships and interactions among various clinical parameters that may not be apparent through traditional modeling techniques. Additionally, deep learning models, such as Convolutional Neural Networks, could potentially integrate diverse types of clinical data, including sequential patient records, to forecast tumor dynamics more accurately, offering a more holistic approach to understanding and predicting patient outcomes post-GKRS [33].

This is why, in our investigation of patients experiencing brain metastases from various primary cancers, we used, in addition to the usual ML algorithms, such as Logistic Regression, Support Vector Machines (SVM), and K nearest neighbors (KNN), tree-based modeling techniques—including Decision Tree, Random Forest, and Boosted decision tree classifiers XGBoost—to anticipate the dynamics of tumors within 3 months following GKRS treatment. Furthermore, we also integrated advanced deep learning models, such as Convolutional Neural Networks (CNNs), to see if we can enhance our predictive accuracy and gain deeper insights into the complex patterns of tumor progression. This approach allowed us to pinpoint essential factors and feature permutations that are crucial in estimating the prognosis for these patients.

## 2. Research Methodology

### 2.1. Overview of Research Design and Participant Details

We outline the framework of our study, detailing the procedures followed and the characteristics of the individuals who participated in our research.

The study protocol retrospectively reviewed the medical records of patients treated with GKRS with mask-based fixation of BMs from various primary cancers, between July 2022 and March 2024 at “Prof. Dr. Nicolae Oblu” Emergency Clinic Hospital—Iasi. All experiments were carried out in accordance with relevant guidelines and regulations. The study used only pre-existing medical data; therefore, patient consent was not required, and since it was retrospective, there was no need for approval from the Ethics Committee of Clinical Emergency Hospital “Prof. Dr. Nicolae Oblu” Iasi.

A total of 77 patients (45 males and 32 females; age range 39 to 85 years old; median age, 64 years old) who were previously diagnosed with BMs were enrolled in this study. General characteristics including age, sex, C1yr—tumor volume at one year control, MTS extra cranial—existence of extra cranial metastases, receiving pretreatment, deceased before 1 year, Karnofsky performance scale (KPS) score [34], number of lesions, beam-on time over the number of isocenters for each of the 3 volumes treated, total tumor volume, and tumor dynamics (progression or regression within 3 months following GKRS treatment), are summarized in Table 1. The study design is shown in Figure 1.

### 2.2. Strategy for Gamma Knife Radiosurgery Implementation

We discuss, in what follows, the systematic approach adopted for administering Gamma Knife radiosurgery (GKRS), focusing on the meticulous planning and execution phases essential for the treatment. All patients underwent GKRS using the Leksell Gamma Knife ICON (Elekta AB, Stockholm, Sweden).

All MRI examinations were performed on a 1.5 Tesla whole-body scanner (GE 174 SIGMA EXPLORER) that was equipped with the standard 16-channel head coil. The MRI 175 study protocol consisted of the following: The conventional anatomical MRI (cMRI) protocol for clinical routine diagnosis of brain tumors, including, among others, an axial fluid-attenuated inversion recovery (FLAIR) sequence as well as a high-resolution contrast-enhanced T1-weighted (CE T1w) sequence.The advanced MRI (advMRI) protocol for clinical routine diagnosis of brain tumors was extended by axial diffusion-weighted imaging (DWI; b values 0 and 1000 s/mm^2^) sequence and a gradient echo dynamic susceptibility contrast (GE-DSC) perfusion MRI sequence, which was performed using 60 dynamic measurements during administration of 0.1 mmol/kg-bodyweight gadoterate meglumine.

All magnetic resonance images were registered with Leksell Gamma Plan (LGP, Version 11.3.2, TMR algorithm), and any images with motion artifacts were excluded. The tumor volumes were calculated by LGP without margin. Generally, the prescription of a total dose of 30 Gy, delivered in 3 stages of GRKS, was selected based on the linear quadratic model [35,36] and the work of Higuchi et al. from 2009 [37]. The GKRS planning was determined through a consensus between the neurosurgeon, radiation oncologist, and expert medical physicist.

### 2.3. Labeling of Medical Data

In the process of labeling medical data, we extracted relevant features from the broad details available in the electronic medical record (EMR) system. One of the prevalent hurdles in machine learning (ML) applications is the issue of incomplete data within the EMR. To tackle this, we applied two principal strategies: imputation and exclusion of data.

Imputation is the method of filling in missing information with plausible values, which proves particularly beneficial when the missing data is scarce. In our research, we opted to replace missing entries in categorical data with the value found in the previous row, whereas for numerical attributes, we used the median value of the dataset. Additionally, for the transformation of categorical variables, label encoding techniques were used [38].

Unbalanced datasets pose challenges in ML when one class vastly outweighs others, which is the case for our study, where only 6 patients in 77 (7.8%) showed signs of progression of lesions. Approaches to address this issue include resampling (oversampling/undersampling), class weighting, cost-sensitive learning, ensemble methods, and data augmentation. The appropriate method depends on the problem and dataset, necessitating evaluation of all classes for accurate results [39].

### 2.4. Data Manipulation Techniques

Our preference for the Python programming language in this study was driven by its simplicity in handling data operations, coupled with its access to a broad spectrum of freely available libraries [40]. The project leveraged the capabilities of open-source Python 3.0 libraries, including NumPy for numerical data manipulation, pandas for data structures and analysis, Matplotlib and seaborn for data visualization, and TensorFlow, Keras, and scikit-learn for machine learning and data mining tasks. 

Logistic Regression is effective in handling binary outcomes, such as the presence or absence of tumor growth. It can utilize various patient and tumor characteristics to predict the probability of specific outcomes, aiding in the risk stratification and management of patients. Support Vector Machine excels in classifying complex patient data into distinct categories, such as predicting the likelihood of tumor recurrence. Its ability to handle high-dimensional data makes it invaluable in analyzing the myriad factors influencing brain metastases, from genetic markers to treatment responses. K Nearest Neighbors offers a straightforward yet powerful method for prognosis by comparing a patient’s data against those of similar patients. This similarity-based approach is especially beneficial in medical cases where individual patient characteristics significantly influence the disease trajectory, allowing for personalized prediction of tumor dynamics. Together, Decision Trees and Random Forest algorithms harness the intricate data landscape of BMs to deliver nuanced and individualized prognostic insights. They enable a structured analysis of the factors driving tumor behavior, facilitating targeted and evidence-based treatment strategies that can lead to better patient management and outcomes [41,42]. 

XGBoost is a machine learning algorithm that belongs to the ensemble learning category, specifically the gradient boosting framework. It uses decision trees as base learners and employs regularization techniques to enhance model generalization. Known for its computational efficiency, feature importance analysis, and handling of missing values, XGBoost is widely used for tasks such as regression, classification, and ranking; therefore, it was a key component of our methodology [43]. To complement the traditional machine learning models, we also incorporated deep learning (DL) techniques using Convolutional Neural Networks (CNNs). These networks can model complex relationships and interactions between the various clinical, demographic, and biological features of our dataset. By employing layers of interconnected neurons, CNNs are capable of capturing intricate patterns and nonlinearities in the data, which enhances our predictive accuracy and provides deeper insights into the factors influencing tumor dynamics. This integration of deep learning into our methodology significantly strengthens our analytical capabilities, enabling more effective risk stratification and personalized treatment strategies. The dataset was randomly divided into training group (*n* = 54) and test group (*n* = 23). We typically adhere to the conventional 80:20 training-to-test set ratio. However, due to the limited volume of data at our disposal, we opted to modify this ratio (70:30) to ensure a more robust evaluation of the test set performance. This adjustment allows for a more comprehensive assessment under constrained data conditions. The input variables were general characteristics. Hyperparameter tuning was conducted using the GridSearchCV function from scikit-learn, targeting the optimization of model parameters. The dependent variable under investigation was tumor dynamics, i.e., progression or regression within 3 months following GKRS treatment, categorized into “Regression” or “Progression”.

To assess the performance of the ML algorithms and DL models, we computed metrics such as accuracy, sensitivity, and specificity, along with the receiver operating characteristic (ROC) curve and the area under the curve (AUC). 

Subsequent to model training, we identify and validate important features and permutation features, with permutation feature importance serving as a versatile technique for evaluating the significance of features across various fitted models, provided the data are structured in tabular form [44].

## 3. Results

### 3.1. Overview and Analysis of Data

The cohort study included 77 patients with BM, focusing on their age at the time of GKRS treatment. The KPS scores were evaluated by physicians. Pathologic diagnoses, statuses of receiving chemotherapy, and pretreatment records were obtained from the EMR. The number of lesions, tumor volume, the number of fractions, and prescription doses were documented in the LGP. Detailed data descriptions are summarized in Table 2. 

Let us interpret the information in the bar charts from Figure 2a. Each chart shows the distribution of patients with tumor regression (labeled ‘0’) versus tumor progression (labeled ‘1’) across different variables within the dataset.

These charts collectively offer valuable insights into factors associated with tumor progression and survival rates in patients. For instance, gender differences and the presence of external metastases are prominently associated with the progression of the tumor. Pre-treatment status is less clear-cut and could be influenced by many factors that the chart does not specify. Finally, the strong correlation between tumor progression and one-year mortality underscores the seriousness of tumor progression as an indicator of patient prognosis. It is important to note that these are correlative relationships, and causation should not be inferred without further, controlled study.

The eight boxplots in Figure 2b show the distribution of various medical variables for two groups of patients: those with tumor regression (labeled ‘0’) and those with tumor progression (labeled ‘1’). In summary, patients with tumor regression generally have lower volumes of tumor at control, fewer metastases, and higher Karnofsky scores, while those with progression show opposite trends. The beam-on time seems to vary less consistently between the groups, with some outliers indicating individual variability in treatment. These boxplots provide a visual summary of how these variables correlate with tumor outcomes in the study population.

### 3.2. Analysis of ML Models

#### 3.2.1. Before Hyperparameter Tuning

Table 3 shows the performance metrics for the six different predictive models on our dataset. Two key performance metrics are reported: Accuracy and the Area under the Receiver Operating Characteristic Curve (AUC).

The Logistic Regression, SVM, Decision Tree, and Random Forest models show nearly identical performance (accuracy and AUC around 0.93), suggesting that the dataset may be too simple or the default settings for these models are similarly effective.

The lower performance (accuracy of 0.8837 and AUC of 0.89) of KNN could be due to its sensitivity to data scale and the default number of neighbors, indicating a need for parameter tuning.

XGBoost is the best performer, with an accuracy of 0.9535 and AUC of 0.95, benefiting from its gradient boosting technique that effectively handles diverse data types. 

The similar high scores across most models might indicate a strong signal in the data or a performance plateau, suggesting that further gains might require hyperparameter tuning or different evaluation metrics, especially in cases of practical application with significant costs associated with prediction errors.

Figure 3 shows the confusion matrices for the six predictive models tested without tuning: Logistic Regression, SVM, KNN, Decision Tree, Random Forest, and XGBoost.

Logistic Regression, SVM, Decision Tree, and Random Forest: these models share an identical confusion matrix with high sensitivity and precision, indicating effective detection of positive cases without any missed positive predictions.

KNN exhibits slightly lower precision due to more false positives but maintains high sensitivity with no missed positives, suggesting it may need parameter adjustments for better precision.

XGBoost shows the best performance with the highest number of true negatives and fewest false positives, making it the most precise and reliable model in this comparison for minimizing incorrect positive predictions.

Overall, all models demonstrate high sensitivity, crucial for applications like medical diagnostics, where missing a positive case can be critical. XGBoost, however, offers superior precision, and the comparable results of the other four models suggest minimal differentiation in performance for this dataset.

The ROC curves in Figure 4 compare the diagnostic ability of the six classification models at various threshold settings. The AUC for each model is a summary measure of the accuracy of the test. Here are the insights based on the provided ROC curves:

SVM and Random Forest achieve perfect AUC scores of 1.00, indicating excellent performance in class separation, but such perfection could suggest overfitting or data issues.

XGBoost is nearly perfect, with an AUC of 0.99, showing exceptional ability to distinguish between classes, slightly behind SVM and Random Forest.

KNN shows a high AUC of 0.98, strong overall performance despite its lower accuracy, and more false positives, highlighting good class differentiation.

Logistic Regression and Decision Tree show solid AUC scores of 0.92 and 0.93, respectively, indicating effective class separation though not as optimal as other models.

Overall, while all models demonstrate strong separability on ROC curves, the unusually high scores for SVM and Random Forest warrant scrutiny for potential overfitting or data anomalies.

The classification report in Table 4 (values in brackets), offers insights into the performance of the six machine learning models without hyperparameter tuning. Here is a detailed comment on each:

Logistic Regression, SVM, Decision Tree, and Random Forest all demonstrate strong, consistent metrics with an accuracy, precision, recall, and F1 score of 0.93, showing balanced performance across classes.

KNN shows lower overall performance with an accuracy of 0.88; it excels in identifying class 1 with perfect recall but struggles with class 0, indicated by lower recall and overall F1 scores.

XGBoost is the top performer, with an accuracy of 0.95; it shows the best recall for class 0 at 0.91 and uniformly high precision and F1 scores across classes, showcasing superior predictive capabilities.

Overall, XGBoost leads in performance metrics, while KNN lags slightly, particularly in recall for class 0. These findings suggest a good baseline for further model refinement and parameter tuning.

#### 3.2.2. After Hyperparameters Tuning

Table 5 presents the accuracy and AUC scores for the six machine learning models after they have undergone hyperparameter tuning. Tuning the models typically involves adjusting various parameters to improve performance.

Logistic Regression and KNN both improved to an accuracy and AUC of 0.95 post-tuning, showing substantial enhancements, especially for KNN.

SVM (Support Vector Machine) emerges as the top performer with the highest post-tuning accuracy (0.9767) and AUC (0.98), indicating highly effective tuning.

Decision Tree shows moderate success with an accuracy of 0.9069 and an AUC of 0.91, the lowest among the models, yet still demonstrating decent predictive ability.

Random Forest: slight performance drop post-tuning to an accuracy of 0.9302 and AUC of 0.93, suggesting better generalization.

XGBoost: unexpected performance decline post-tuning, with the lowest accuracy (0.8837) and AUC (0.89), possibly due to suboptimal tuning or overfitting.

Overall, tuning had mixed effects; while SVM, Logistic Regression, and KNN showed improvement or maintained high metrics, Random Forest slightly declined, and XGBoost significantly underperformed, underscoring the critical nature of precise tuning strategies.

Figure 5 shows the confusion matrices for the six models tested after hyperparameter tuning. 

Logistic Regression and KNN: both models show strong performance with 20 and 21 true negatives, respectively, and both correctly predict all true positives (21) with only a few false positives (2 each), indicating high precision and sensitivity.

SVM (Support Vector Machine) achieves the best performance, with 21 true negatives, 21 true positives, and the lowest amount of false positives (1), showing optimal precision and no false negatives.

Decision Tree and Random Forest: both models perform well with 18 and 19 true negatives, respectively, and complete accuracy in predicting true positives (21), though Decision Tree has more false positives (4) compared to Random Forest (3).

XGBoost experiences a slight decline, with 18 true negatives, 20 true positives, a higher amount of false positives (4), and a small number of false negatives (1), indicating a decrease in both precision and sensitivity.

Overall, post-tuning results display generally high performance across models, particularly highlighting SVM’s exemplary precision and accuracy, while XGBoost shows a minor reduction in performance metrics, emphasizing the importance of careful tuning and model selection based on the application’s critical needs.

The provided Figure 6 shows the ROC curves for the six models after hyperparameter tuning.

SVM and Random Forest: both achieve perfect AUC scores of 1.00, suggesting ideal classification but raising concerns about possible overfitting or data leakage.

Logistic Regression shows strong separability, with an AUC of 0.95, indicating robust performance.

KNN: significant improvement post-tuning with an AUC of 0.98, reflecting high class separation.

Decision Tree: good performance, with an AUC of 0.93, though slightly lower than other models, potentially due to inherent model characteristics.

XGBoost: despite lower precision in other metrics, the AUC of 0.98 indicates excellent overall ability to rank classes correctly.

Overall, tuning has enhanced the models’ discriminative capabilities, though the perfect scores of SVM and Random Forest warrant further investigation to ensure they are not artifacts of overfitting. The AUC is especially valuable in evaluating models dealing with imbalanced datasets.

The provided classification report in Table 4 after hyperparameter tuning (values outside the brackets) shows how parameter adjustments have influenced the performance metrics of the six machine learning models. Let us analyze the performance of each model after tuning:

Logistic Regression shows significant improvements in all metrics post-tuning, notably in accuracy and recall, achieving an F1-score of 0.95 for both classes.

SVM: major gains across the board, with accuracy rising to 0.98 and high F1 scores, positioning it as the top performer post-tuning.

KNN: marked improvements in accuracy (0.88 to 0.95) and recall for class 0 (0.77 to 0.91), significantly boosting its overall performance.

Decision Tree: slight drop in performance, with reduced accuracy and recall for class 0, indicating a decrease in effectiveness.

Random Forest: consistent improvements similar to Logistic Regression, with a steady F1 score of 0.95 and better precision and recall.

XGBoost: unexpected decline post-tuning, with reduced accuracy and a significant drop in precision for class 0, suggesting less optimal tuning.

Overall, while SVM and KNN exhibited substantial improvements, making them highly effective post-tuning, Decision Tree and XGBoost experienced performance dips, highlighting the critical need for careful hyperparameter tuning and model validation.

In light of this analysis, while SVM shows high accuracy and reliability post-tuning, XGBoost offers robust initial performance. The choice between these models may depend on specific clinical needs and data characteristics. For settings where interpretability and initial robust performance are crucial, XGBoost may be preferable, while SVM could be better suited for scenarios requiring high precision.

### 3.3. Convolutional Neural Network Model Analysis

Model Architecture
-Type: 1D Convolutional Neural Network (CNN).-Layers:
1.Conv1D Layer: 64 filters, kernel size of 2, ReLU activation; processes sequential data by extracting local feature patterns.2.MaxPooling1D Layer: pool size of 2; reduces the dimensionality of the data, which helps in reducing overfitting.3.Flatten Layer: converts the pooled feature map into a 1D array to prepare data for dense layers.4.Dense Layer: 50 neurons, ReLU activation; introduces non-linearity and learns complex patterns.5.Output Layer: one neuron, sigmoid activation; provides binary classification output on the probability scale.-Compilation: uses binary crossentropy loss and an Adam optimizer, and tracks accuracy metrics.Training Performance
-Accuracy: achieved high training and validation accuracy, showing stable performance over epochs. The model’s test accuracy closely follows training accuracy, indicating good generalization.-Loss: loss decreases sharply initially and then gradually levels off, which is typical and desirable in a training scenario. Both training and validation loss converge, suggesting minimal overfitting (see Figure 7).Confusion Matrix Results
-True Positives: 21-True Negatives: 21-False Positives: 1-False Negatives: 0-Excellent performance in accurately classifying both positive and negative cases (see Figure 8).
ROC Curve Analysis
-AUC: 0.97-Demonstrates excellent discrimination capability between the classes. The curve quickly rises to a high true positive rate at a very low false positive rate (see Figure 9).Classification Report Results (see Table 6)
-Precision: 0.95 (Class 1), 1.00 (Class 0)-Recall: 1.00 (Class 1), 0.95 (Class 0)-F1 Score: 0.98 for both classes-Overall Accuracy: 0.98-Consistently high metrics across all classes indicate a balanced and effective model.

To summarize, the 1D CNN has demonstrated outstanding predictive performance, validated through high accuracy, precision, recall, and F1 scores across different metrics and visual evaluations like ROC and the confusion matrix. This model is robust, with great potential for deployment in practical settings where accurate and reliable predictions are critical. 

The 1D CNN model provides a robust alternative to traditional machine learning models, offering comparable if not superior accuracy and reliability for predictive tasks in medical imaging and diagnosis. Its ability to perform well without extensive tuning suggests a strong adaptive capacity, making it a valuable tool for clinical applications where rapid and accurate predictions are essential. However, the choice between a 1D CNN and traditional models would ultimately depend on the specific requirements of the application, including the need for model interpretability, computational resources, and expertise in model training and deployment. Future steps could include further validation with external datasets, potential application in broader contexts, and continuous monitoring to maintain performance in dynamic real-world conditions.

### 3.4. Assessment of Feature Variables

We gave the features detailed in Table 1 and Table 2, for the sake of simplicity, the following numbers: 0—Age, 1—Sex, 2—C1yr, 3—Mts_ext, 4—T_sist, 5—Dec_1yr, 6—Karn, 7—Nr_Mts, 8—B_on1, 9—B_on2, 10—B_on3, and 11—Vol_tum.

The four diagrams in Figure 7 represent the feature importances as calculated by four different models: Logistic Regression, Decision Tree, Random Forest, and XGBoost. Let us comment on each:

In our analysis of different machine learning models, each uses distinct methods to assess the influence of features on predictions (see Figure 10): 

Logistic Regression: Coefficients indicate the direction and strength of a feature’s influence on prediction outcomes. For example, ‘Dec_1yr’ and ‘Sex’ negatively affect the predictions, decreasing the probability of a ‘1’ outcome, while ‘C1yr’ has a positive effect, increasing it.

Decision Tree and Random Forest: These models evaluate the importance of features based on their ability to improve the purity of splits, but do not indicate the direction of their influence. In the Decision Tree, ‘Sex’ and ‘Age’ are key, while in the Random Forest, ‘C1yr’ is most crucial, followed by ‘B_on2’ and ‘B_on3’.

XGBoost: Similar to Random Forest in prioritizing ‘C1yr’, but differs in the secondary features’ importance, highlighting ‘Age’ and ‘B_on1’. This model corrects errors progressively as trees are added, emphasizing how features contribute to model improvements.

Overall, ‘C1yr’ emerges as a consistently significant predictor across models, underscoring its strong predictive relationship with the target variable. However, the interpretation of feature importance varies by model—Logistic Regression considers the direction of influence (positive or negative), while tree-based models focus on how effectively features separate the data, a measure of node purity. Each model’s methodology and assumptions should be considered when interpreting these importances within the specific context of their application, keeping in mind that statistical significance does not necessarily equate to practical relevance.

## 4. Discussion

This study was designed to investigate the prediction of tumor dynamics within 3 months after GKRS for BM patients. The main findings of this study are as follows: XGBoost significantly outperformed other models with an accuracy of 0.9535 and an AUC of 0.95 before tuning. Post-tuning, the Support Vector Machine (SVM) demonstrated the most substantial improvement, achieving an accuracy of 0.9767 and an AUC of 0.98. The 1D CNN model achieved excellent performance with an accuracy of 98%, an AUC of 0.97, and consistently high precision, recall, and F1 scores across classes, demonstrating its robustness and reliability in accurately classifying outcomes. Important features are “control over one year”, “age of the patient”, and “beam-on time on V1”. 

ML algorithms are increasingly being applied to predict treatment outcomes for patients with brain metastases undergoing GKRS. These algorithms can analyze vast datasets from medical records, imaging studies, and treatment parameters to identify patterns and predict tumor response, patient survival, and risk of complications. For example, ML models have been developed to predict overall survival, local control, and radiation necrosis in patients treated with GKRS for brain metastases. These predictive models can aid clinicians in selecting patients who are most likely to benefit from GKRS, thereby optimizing individual patient outcomes.

The 2016 study by Sneed et al., published in the “International Journal of Radiation Oncology”, showcased the use of machine learning (ML) algorithms to accurately predict the recurrence of brain metastases after Gamma Knife radiosurgery (GKRS), utilizing clinical and treatment variables [45]. This technological advance allows for tailored follow-up and treatment modifications, enhancing both survival and quality of life. Extending beyond Sneed’s research, Smith et al. (2018) employed deep learning algorithms to predict survival rates after GKRS, demonstrating superior predictive power over traditional models [46].

Zhou et al. (2019) further explored the potential of ML in predicting severe side effects such as radiation necrosis from GKRS, using non-invasive methods to enhance patient safety [47]. The predictive capabilities of ML in these studies exemplify its role in refining prognostic assessments and improving clinical outcomes.

Subsequent research by El Naqa et al. (2017) and Gupta et al. (2020) has focused on utilizing ML to optimize and automate GKRS treatment planning. These studies highlight how ML algorithms can enhance treatment precision by recommending optimal radiation doses and angles, thus improving the efficiency and accuracy of treatments [48,49].

Furthermore, Liu et al. (2021) demonstrated the application of ML in determining personalized treatment margins for GKRS, significantly reducing the risk to adjacent healthy tissues and optimizing therapeutic outcomes [50]. Contributions such as these underscore the transformative impact of ML in advancing personalized medicine within oncology.

Additional innovative work includes that of Mayinger et al. (2020) and Kessler et al. (2021), who have applied ML to tailor treatment plans based on genetic and molecular profiles, thereby significantly enhancing therapeutic effectiveness [51,52]. These advancements pave the way for treatments that are not only more precise but also more responsive to the individual characteristics of each patient’s disease.

Chang et al. (2022) further expanded the use of ML in guiding the selection of adjuvant therapies post-GKRS, analyzing patient data to predict which individuals would benefit from additional interventions like chemotherapy or immunotherapy [53]. This approach highlights a significant shift towards a more nuanced and personalized approach to cancer treatment, aiming to maximize benefits and minimize risks for patients. Despite the promising advances, integrating ML into the clinical workflow for treating brain metastases with GKRS faces several challenges. These include the need for large, high-quality datasets to train and validate ML models, addressing data privacy and security concerns, and ensuring the interpretability of ML algorithms for clinical decision making. Moreover, the clinical implementation of ML requires a multidisciplinary approach, involving oncologists, radiologists, data scientists, and IT professionals to ensure seamless integration into the healthcare system.

As we move forward, continuous research and development in ML algorithms and Deep Learning (DL), along with advancements in computational power and data analy-tics, are expected to further enhance the treatment of brain metastases with GKRS. The integration of ML and DL offers a promising avenue for improving outcomes through personalized treatment plans, predictive analytics, and refined decision-making processes. Future studies focusing on the validation of ML models and Deep Learning in clinical trials, and their implementation in routine clinical practice will be critical to realizing the full potential of ML and DL in this field.

In this study, we face some limitations. The number of patients with BMs was relatively small, and all data were collected at “Prof. Dr. Nicolae Oblu” Emergency Clinic Hospital—Iasi; thus, our model could be prone for overfitting to our hospital and we need to incorporate multicenter data in the future studies. 

## 5. Conclusions

In this study, we analyzed tumor responses over three months post-Gamma Knife radiosurgery (GKRS) in 77 patients with brain metastases. We conducted a comprehensive evaluation of six machine learning models to gauge performance metrics such as accuracy, AUC, and other indicators derived from confusion matrices, both pre- and post-hyperparameter tuning.

Initially, the XGBoost model excelled above the rest, achieving an accuracy of 0.9535 and an AUC of 0.95, showcasing its effectiveness in handling the dataset. Comparable results were observed with Logistic Regression, SVM, Decision Tree, and Random Forest, each posting an accuracy and AUC of around 0.93, indicating minimal variance in performance across these models for this specific dataset. However, the KNN (K-Nearest Neighbors) model underperformed, with lower accuracy and AUC scores, underscoring the necessity for precise parameter adjustments.

Post-tuning, the SVM model displayed the most significant improvement, leading with an accuracy of 0.9767 and an AUC of 0.98. Logistic Regression and KNN also showed substantial enhancements, each reaching an accuracy and AUC of 0.95. In contrast, Decision Tree and XGBoost saw declines in performance after tuning, possibly due to overfitting or inadequate parameter settings.

The 1D CNN model demonstrated superior classification abilities, achieving an overall accuracy of 98% and an impressive AUC of 0.97, effectively distinguishing between positive and negative classes. The model also recorded high precision and recall values—0.95 and 1.00 for the positive class, respectively—and perfect precision for the negative class, with F1 scores of 0.98 for both classes, confirming its balanced performance across precision and sensitivity. These metrics underscore the model’s consistent and reliable predictive accuracy.

Our analysis also delved into feature importance, with ‘C1yr’ emerging as a pivotal predictor across all models. The approach to interpreting feature importance varied, from direct impacts on outcomes in Logistic Regression to assessing data split quality in tree-based models.

This research pioneers the examination of tumor dynamics following staged GKRS, indicating no significant differences in outcomes across various protocols for treating larger tumors, such as fractionated GKRS [54]. We introduced innovative considerations of beam-on time and intervals between fractions on treatment efficacy. With the growing prevalence of fractionation schemes, these factors are likely crucial in enhancing treatment effectiveness. Future studies should aim to validate these protocols against tumor volume and patient characteristics to determine the most effective strategies for achieving optimal clinical outcomes.

## Figures and Tables

**Figure 1 diagnostics-14-01268-f001:**
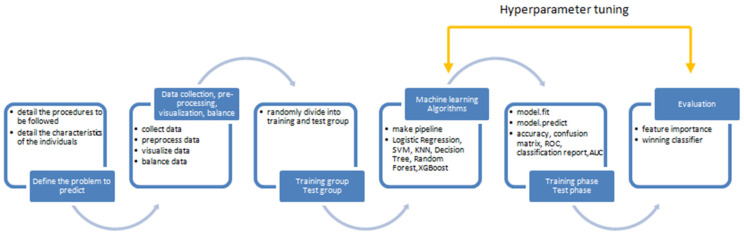
Schematic of the machine learning deep learning study design.

**Figure 2 diagnostics-14-01268-f002:**
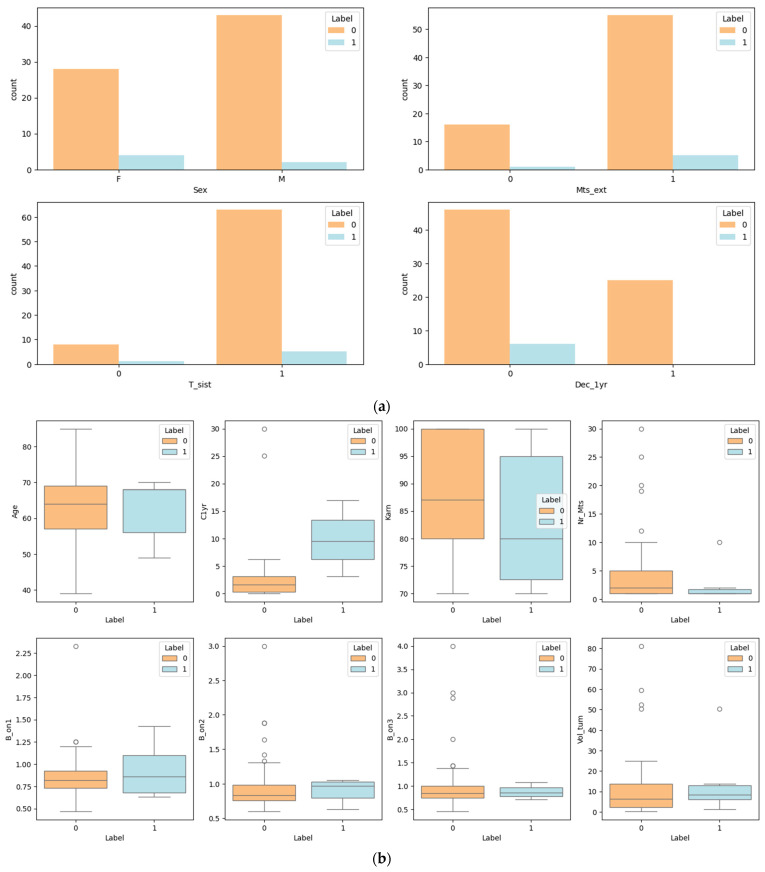
(**a**) Bar plot for categorical features versus target; (**b**) box plot for numerical features versus target.

**Figure 3 diagnostics-14-01268-f003:**
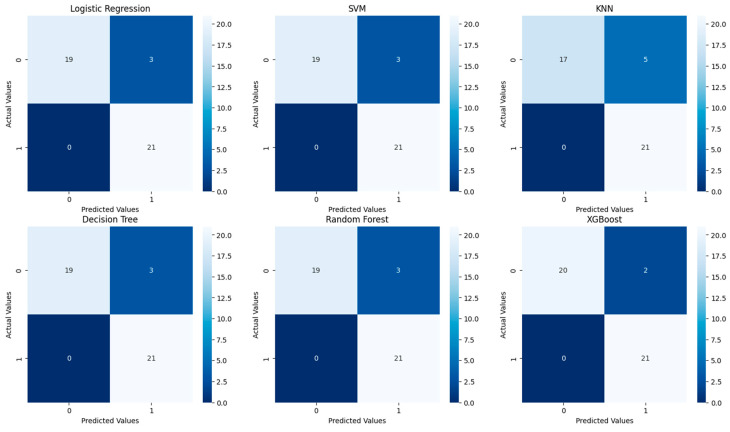
Confusion matrices for the six models tested without tuning.

**Figure 4 diagnostics-14-01268-f004:**
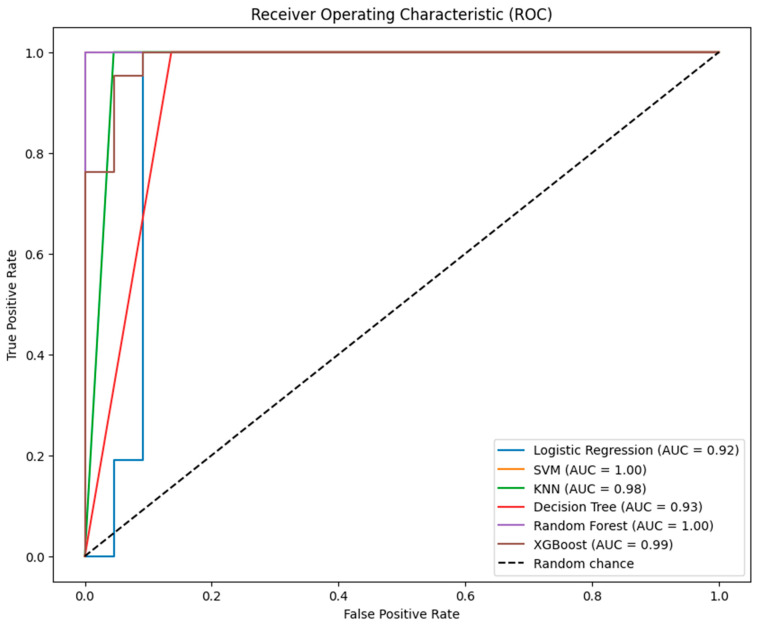
ROC curves for the six models tested without tuning.

**Figure 5 diagnostics-14-01268-f005:**
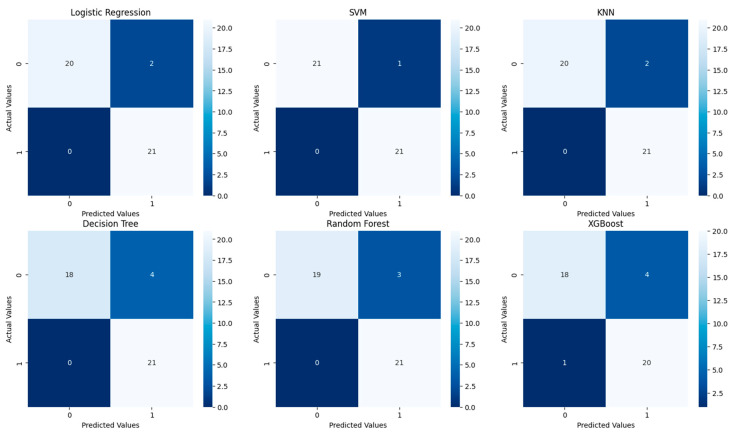
Confusion matrices for the six models tested after tuning.

**Figure 6 diagnostics-14-01268-f006:**
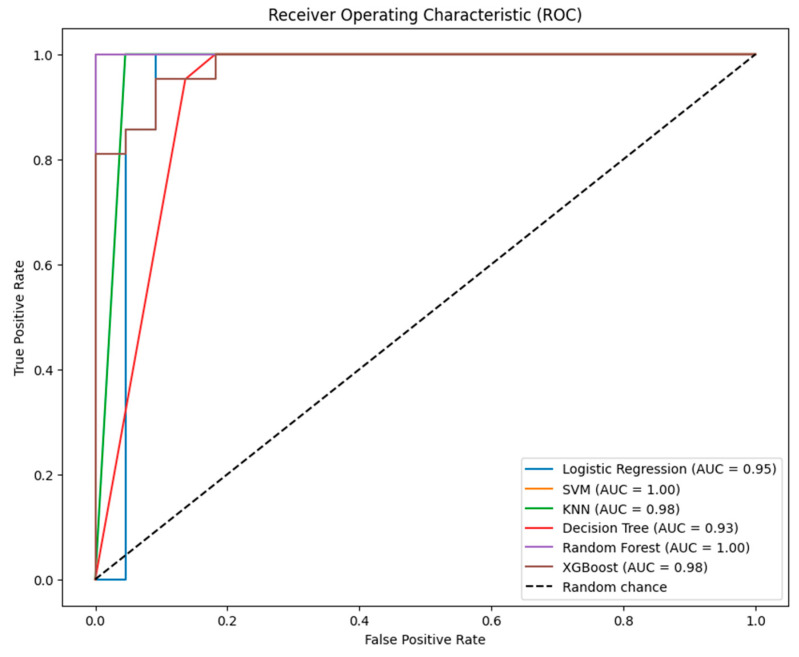
ROC curves for the six models tested after tuning.

**Figure 7 diagnostics-14-01268-f007:**
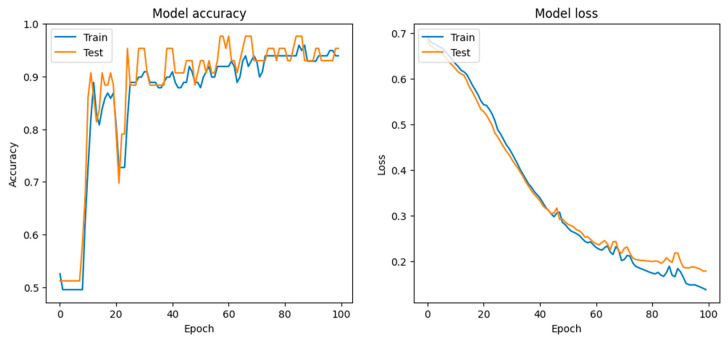
Model’s accuracy and loss during training.

**Figure 8 diagnostics-14-01268-f008:**
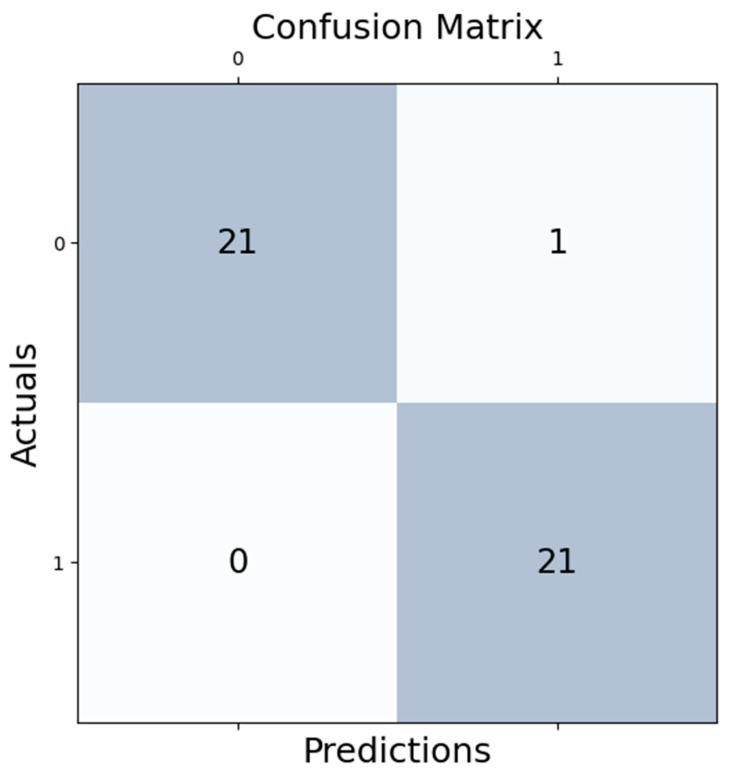
Confusion matrix for the CNN model used.

**Figure 9 diagnostics-14-01268-f009:**
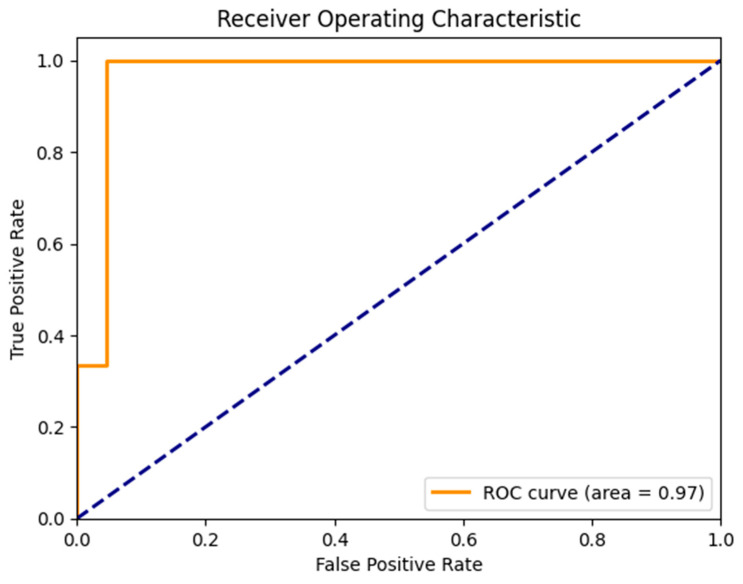
ROC curve for the CNN model used.

**Figure 10 diagnostics-14-01268-f010:**
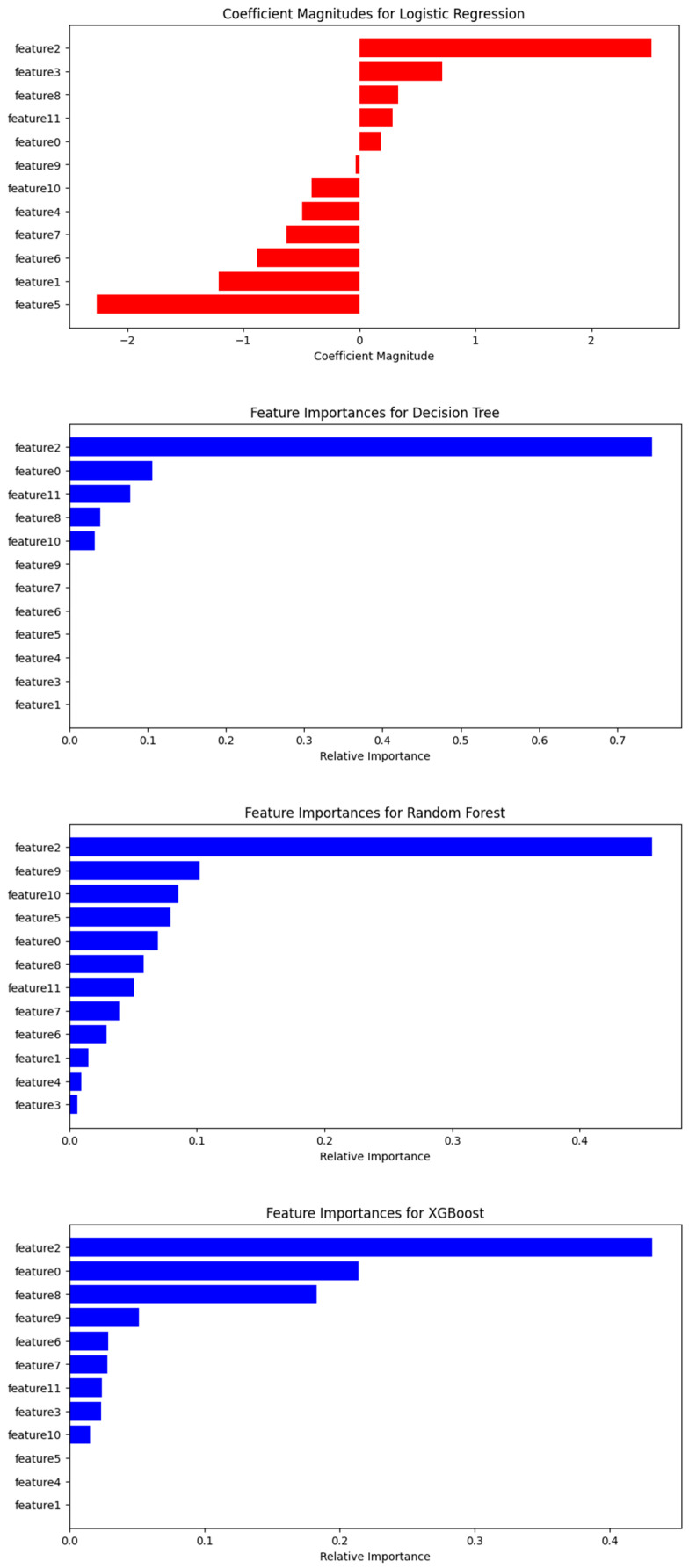
Feature importance according to Logistic Regression, Decision Tree, Random Forest, and XGBoost (no direct feature importance plot available for: SVM and KNN). See Table 2 for feature number correspondences.

**Table 1 diagnostics-14-01268-t001:** Patient demographics for brain metastases from various primary cancers.

Characteristics	Value
Number of patients	77
Age (yr)	
Median (range)	64 (39–85)
Sex	
Male (%)	45 (58.44%)
Female (%)	32 (41.56%)
C1yr—control over one year (cm^3^)	17 missing data
Median(range)	0.9 (0–30)
Patients with extra cranial MTS	6 missing data
	54
Receiving pre-treatment, systemic treatment	68
Deceased before 1 year	1 missing data
	25
KPS score	9 missing data (11.69%)
100	26 (33.77%)
90	8 (10.39%)
80	22 (28.57%)
70	12 (15.58%)
The number of lesions	
Median (range)	2 (1–30)
1–3	52
4–6	12
7–10	8
>10	5
Beamon time on V1 (min/cm^3^)	
Median (range)	0.82 (0.47–2.33)
Beamon time on V2 (min/cm^3^)	1 missing data
Median (range)	0.83 (0.60–3.00)
Beam-on time on V3 (min/cm^3^)	2 missing data
Median (range)	0.83 (0.46–4.00)
Total tumor volume (# of patients with):	
<5 cm^3^	34
≤10 cm^3^	13
>10 cm^3^	30
Tumor dynamics (# of patients with):	
- Progression	6
- Regression	71

**Table 2 diagnostics-14-01268-t002:** Summary of features categorized for machine learning algorithms.

Feature	Feature Number	Description	Categorical/Numeric Data for Machine Learning Algorithm	Type
Age	0	Age at time of treated GKRS	No categorical	Discrete
Sex	1	Biological sex	0 = Female; 1 = Male Label encoding	Numeric
C1yr—control over one year	2	Volume of lesion measured at the 1 year control	No categorical	Numeric
Patients with extra cranial MTS	3	Patients having detected with extracranial metastases	0 = No; 1 = Yes Label encoding	Numeric
Receiving pre-treatment	4	Before GKRS, treated by surgery or radiotherapy, or performed chemotherapy	0 = No pretreatment; 1 = Pretreatment Label encoding	Numeric
Deceased before 1 year	5	Patients who passed away before 1 year after receiving GKRS	0 = Alive; 1 = Deceased Label encoding	Numeric
KPS score	6	KPS score runs from 0 to 100. Three physicians allow to evaluate the patient ability to receive GKS for BM.	No categorical	Numeric
The number of lesions	7	This divided the 4 groups based on the number of lesions.	No categorical	Discrete
Beam-on time on V1	8	The beam-on time on V1 treated over the number of isocenters in V1	No categorical	Numeric
Beam-on time on V2	9	The beam-on time on V2 treated over the number of isocenters in V2	No categorical	Numeric
Beam-on time on V3	10	The beam-on time on V3 treated over the number of isocenters in V3	No categorical	Numeric
Total tumor volume	11	This divided the 3 groups based on the number of volumes.	No categorical	Numeric
Tumor dynamics	Label	Tumor progression or regression within 3 months following GKRS treatment.	0 = Regression; 1 = Progression	Discrete

**Table 3 diagnostics-14-01268-t003:** Accuracy and AUC for the six models tested without tuning.

Nr.	Model	Accuracy	AUC
1	Logistic Regression	0.9302	0.93
2	SVM	0.9302	0.93
3	KNN	0.8837	0.89
4	Decision Tree	0.9302	0.93
5	Random Forest	0.9302	0.93
6	XGBoost	0.9535	0.95

**Table 4 diagnostics-14-01268-t004:** Classification report for the six models tested without tuning (values in brackets) and with tuning (values outside the brackets).

Classification Report for Logistic Regression				
	precision	recall	F1 score	Support
0	(1.00) 1.00	(0.86) 0.91	(0.93) 0.95	22
1	(0.88) 0.91	(1.00) 1.00	(0.93) 0.95	21
accuracy			(0.93) 0.95	43
macro avg	(0.94) 0.96	(0.93) 0.95	(0.93) 0.95	43
weighted avg	(0.94) 0.96	(0.93) 0.95	(0.93) 0.95	43
**Classification Report for SVM**				
	precision	recall	F1 score	Support
0	(1.00) 1.00	(0.86) 0.95	(0.93) 0.98	22
1	(0.88) 0.95	(1.00) 1.00	(0.93) 0.98	21
accuracy			(0.93) 0.98	43
macro avg	(0.94) 0.98	(0.93) 0.98	(0.93) 0.98	43
weighted avg	(0.94) 0.98	(0.93) 0.98	(0.93) 0.98	43
**Classification Report for KNN**				
	precision	recall	F1 score	Support
0	(1.00) 1.00	(0.77) 0.91	(0.87) 0.95	22
1	(0.81) 0.91	(1.00) 1.00	(0.89) 0.95	21
accuracy			(0.88) 0.95	43
macro avg	(0.90) 0.96	(0.89) 0.95	(0.88) 0.95	43
weighted avg	(0.91) 0.96	(0.88) 0.95	(0.88) 0.95	43
**Classification Report for Decision Tree**				
	precision	recall	F1 score	Support
0	(1.00) 1.00	(0.86) 0.82	(0.93) 0.90	22
1	(0.88) 0.84	(1.00) 1.00	(0.93) 0.91	21
accuracy			(0.93) 0.91	43
macro avg	(0.94) 0.92	(0.93) 0.91	(0.93) 0.91	43
weighted avg	(0.94) 0.92	(0.93) 0.91	(0.93) 0.91	43
**Classification Report for Random Forest**				
	precision	recall	F1 score	Support
0	(1.00) 1.00	(0.86) 0.91	(0.93) 0.95	22
1	(0.88) 0.91	(1.00) 1.00	(0.93) 0.95	21
accuracy			(0.93) 0.95	43
macro avg	(0.94) 0.96	(0.93) 0.95	(0.93) 0.95	43
weighted avg	(0.94) 0.96	(0.93) 0.95	(0.93) 0.95	43
**Classification Report for XGBoost**				
	precision	recall	F1 score	Support
0	(1.00) 0.95	(0.91) 0.82	(0.95) 0.88	22
1	(0.91) 0.83	(1.00) 0.95	(0.95) 0.89	21
Accuracy			(0.95) 0.88	43
macro avg	(0.96) 0.89	(0.95) 0.89	(0.95) 0.88	43
weighted avg	(0.96) 0.89	(0.95) 0.88	(0.95) 0.88	43

**Table 5 diagnostics-14-01268-t005:** Accuracy and AUC for the six models tested after tuning.

Nr.	Model	Accuracy	AUC
1	Logistic Regression	0.9535	0.95
2	SVM	0.9767	0.98
3	KNN	0.9535	0.95
4	Decision Tree	0.9070	0.91
5	Random Forest	0.9302	0.93
6	XGBoost	0.8837	0.89

**Table 6 diagnostics-14-01268-t006:** Classification report for the CNN model.

Classification Report for CNN				
	precision	recall	F1 score	Support
0	1.00	0.95	0.98	22
1	0.95	1.00	0.98	21
Accuracy			0.98	43
macro avg	0.98	0.98	0.98	43
weighted avg	0.98	0.98	0.98	43

## Data Availability

No new data were created or analyzed in this study. Data sharing is not applicable to this article.

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
