# Peer review of "Predicting Tumor Dynamics Post-Staged GKRS: Machine Learning Models in Brain Metastases Prognosis"

_diagnostics, 2024, doi:10.3390/diagnostics14121268_

Round 1

Reviewer 1 Report

Comments and Suggestions for Authors

Dear authors,

This study evaluates the predictive performance of six machine learning models in forecasting tumor dynamics within three months following Gamma Knife Radiosurgery in 77 brain metastasis patients. It assesses each model before and after hyperparameter tuning using metrics like accuracy and AUC.

However, post-tuning, the Support Vector Machine (SVM) showed the most significant improvement, achieving an accuracy and AUC of 0.98, while XGBoost's performance declined, indicating potential overfitting. The study also highlights the consistent importance of features. This comprehensive evaluation emphasizes the importance of model selection and hyperparameter tuning in medical diagnostics, where precise positive predictions are crucial.

As the authors rightly point out decisions are often based on expert consensus and guidelines in clinical settings, especially when evidence is limited. Simplified information structures aid in effectively modelling observations and drawing conclusions. Machine learning (ML) techniques are valuable for their ability to learn from data autonomously. Decision trees are beneficial for their clear and accurate classification, which is vital in healthcare.

Of course, there is always the problem that small datasets can lead to overfitting, so methods like random forests and boosted decision trees improve prediction accuracy by exploring variable relationships.

In this study about brain metastasis post-GKRS, authors used tree-based models such as Decision Trees, Random Forests, and XGBoost, along with Logistic Regression, SVM, and KNN. This is indeed a comprehensive theory to identify critical factors and combinations of characteristics necessary to predict patient outcomes.

The findings of this research lead to the conclusion that the k-Nearest Neighbor classifier stands out as the most effective choice for classifying vertebral column diseases, consistently achieving results exceeding 90% across all evaluated metrics. Conversely, the Naïve-Bayes classifier emerges as the least reliable option, displaying a higher number of misclassified disease elements.

The manuscript shows introductory background material sufficient for someone not an expert in this area to understand the context and significance of this work, with good references to follow. The introduction section in particular is very well written and very detailed to create a complete picture to the reader of how to approach all problems with the machine learning method.

We all agree that the ultimate goal of incorporating machine learning (ML) algorithms into the treatment of patients with brain metastases is to advance personalized medicine. By leveraging patient-specific data, ML models can predict individual responses to Gamma Knife Radiosurgery (GKRS), identify potential side effects, and recommend personalized treatment plans that optimize efficacy while minimizing adverse outcomes. These sophisticated models analyze vast amounts of clinical, genetic, and imaging data to provide insights that are beyond human capability. Consequently, this approach not only enhances the precision of treatment but also improves overall patient outcomes by tailoring interventions to the unique characteristics of each patient.

Furthermore, the integration of ML algorithms in clinical practice allows for continuous learning and adaptation. It is very important to conclude by the researchers that XGBoost outperformed all other models with the highest accuracy (0.9535) and AUC (0.95), indicating its robustness in handling the used data set. But finally, after tuning SVM appears most reliable, significantly reducing false positives and maintaining high true positive rates. So overall which of the two would you recommend and why?

I think the downside of having a lot of data to properly train the algorithm I think remains. The conclusions are very important, but they should also be verified by clinical practice to be used more widely. Until then they can clearly act as a direction, until they are verified by other publications with a larger number of participants. Of course, this does not reduce the value of this research at all. In addition, it is very important that the features “control over one year”, “age of the patient”, and “beam on time on V1” are highlighted. More emphasis should be placed there on subsequent research.

In conclusion, based on this research, we can conclude that the final conclusion is that future research should focus on validating treatment protocols based on tumor volume and patient characteristics to ascertain the most effective strategies to achieve optimal clinical outcomes, as formulated I believe that is the next stage of this research. However, confirmation of the dynamics in staged GKRS and the interaction of beam time and intrafraction interval as to whether they affect treatment efficacy should be preceded by much larger samples. This is a key limitation of this research. I would be happy to see the continuation of this research as described by the authors.

Turnitin returned a similarity index of 31% with the bibliography and 21% without a bibliography and quotes, so it’s marginally OK, according to Turnitin didn’t show very high plagiarism. At the same time, the AI writing report contains the total percentage of prose sentences contained in a long form of writing within the submitted document that Turnitin's model identifies as AI-generated as 8%.

Specific comments

1)      Please improve the appearance of Table 5.

2)      Please rephrase the: “We provide first, an exhaustive analysis of six machine…”, in the discussion section.

3)      It is very important to conclude by the researchers that XGBoost outperformed all other models with the highest accuracy (0.9535) and AUC (0.95), indicating its robustness in handling the used data set. How do you explain that XGBoost, despite its strong start, showed reduced performance after tuning with the lowest accuracy (0.8837) and AUC (0.89) among the models. Does this unexpected result suggest to you only that the tuning may not have been optimal, or that the model overfit its training data?

Comments on the Quality of English Language

Overall, the English in this paper is very good.

Reviewer 2 Report

Comments and Suggestions for Authors

Major Comment

1.       The use of decision trees, random forests, SVM, KNN, etc. proves no novelty. These ML models have been in use for decades. The most recent of the model RF dates back to 2000/2001. This is a significant drawback as it relies on older machine learning models. While models like Logistic Regression, Decision Trees, and Random Forests are well-established and robust, they do not leverage the latest advancements in machine learning, such as deep learning models and newer ensemble methods which have shown superior performance in many tasks.

2.       Major revisions should include the consideration and implementation of more recent machine learning models and techniques to enhance the study's robustness and relevance.

Other comments

3.      Include the practical implications of the findings and any noted limitations of the study in the abstract section.

4.      The statement “The out-look for NSCLC patients facing BM remains bleak, with approximately 10% dying within two months following their diagnosis [21,22]” is an old statistic. The reference dates back to 2003/2004.

5.      The statement “Beyond lung cancer, other cancers such as breast, colorectal, prostate, ovarian, and renal cancers also frequently metastasize to the brain, significantly impacting patient outcomes and treatment approaches.” is obtained from????? Please reference.

6.      There are lots of outdated references cited. References older than 10-15 should be avoided as a standard. Older references may be allowed in cases such as historical references.

Comments on the Quality of English Language

Moderate editing of English language required

Reviewer 3 Report

Comments and Suggestions for Authors

The manuscript needs major rework for being reviewed:

-it is far too long

-results are being presented in the methods section

-data is scattered all over the manuscript

-the conclusion section is one page long

-it is quite impossible to read in the current format

So, as this manuscript looks like an abbreviated PhD thesis, I will humbly ask the authors to convert it to an article and resubmit. 

Comments on the Quality of English Language

English proofreading is needed throughout. 

Round 2

Reviewer 2 Report

Comments and Suggestions for Authors

Good job.

Please address the following minor comments:

  1. Why do the metrics in Figure 4 have different values, e.g., the classification report for logistic regression shows a recall of 0.86 and 0.91? This is applicable to all other metrics as well.

  2. The data used seems imbalanced. There are 22 instances for class 0 and 21 instances for class 1. In an imbalanced dataset, metrics like accuracy can be misleading.

Reviewer 3 Report

Comments and Suggestions for Authors

The manuscript has been obviously improved. 
